# Shellac: From Isolation to Modification and Its Untapped Potential in the Packaging Application

**Arihant Ahuja and Vibhore Kumar Rastogi \***

Department of Paper Technology, Indian Institute of Technology Roorkee, Roorkee 247667, Uttarakhand, India
* Correspondence: vibhore.rastogi@pt.iitr.ac.in; Tel.: +91-1322714338

**Abstract:** Recently, terms such as sustainable, bio-based, biodegradable, non-toxic, or environment-benign are being found in the literature, suggesting an increase in green materials for various applications in the future, particularly in the packaging application. The unavoidable shift from conventional polymers to green materials is difficult, as most bio-sourced materials are not water-resistant. Nonetheless, Shellac, a water-resistant resin secreted by a lac insect, used as a varnish coat, has been underutilized for packaging applications. Here, we review Shellac's potential in the packaging application to replace conventional polymers and biopolymers. We also discuss Shellac's isolation, starting from the lac insect and its conversion to Sticklac, Seedlac, and Shellac. Further, the chemistry of shellac resin, the chemical structure, and its properties are examined in detail. One disadvantage of Shellac is that it becomes stiff over time. To enable the usage of Shellac for an extended time in the packaging application, a modification of Shellac via physical and chemical means is conferred. Furthermore, the usage of Shellac in other polymer matrices and its effect are reviewed. Lastly, the non-toxic and biodegradable nature of Shellac and its potential in packaging are explored by comparing it with traditional crude-based polymers and conventional bio-based materials.

**Keywords:** Shellac; lac resin; edible packaging; biodegradable packaging

## 1. Introduction

The history of synthetic polymers started in 1838 by modifying cellulose with camphor, which was used as a substitute for ivory [1–3]. Later, in 1920, the existence of macromolecules was proved in the world of material science by Hermann Staudinger, also known as the father of polymer science [4,5]. The continued innovation in polymers has led to the development of versatile polymers with different properties to ease our lifestyles. In packaging applications, polymers dominate over other traditional materials, such as metals, glass, and ceramics [6]. However, in the search for polymer, we are continuously downgrading the environment. To date, we have produced 8.3 billion tonnes of plastic since the 1950s [7], and about 500 billion single-use plastic cups are used yearly [8], most of them non-biodegradable and usually landfilled. In 2018, more than 50% of plastic wastage was produced by packaging materials [9]. These packaging materials are either mono-material or multi-material, in the form of rigid or flexible packaging, making them difficult to recycle. Many countries claim to recycle polymers and pledge to increase the recycling capacity, but 91% of the plastic has not yet been recycled. Hence, the opportunities to find an alternative to non-biodegradable polymers have surged in the past few years, driven by renewability, circular economy, environmental concern, and health issues [10].

Biopolymers, bio-based polymers, and biodegradable polymers are the most researched packaging materials (Figure 1). Most of these biopolymers have failed in the industrial and consumer markets due to drawbacks, such as their hygroscopic nature [11], water solubility [12], low barrier, and low heat deflection temperatures [13], rendering them unsuitable for food packaging applications. However, they have several benefits over traditional polymers, including renewability, abundance, non-toxicity, and biocompatibility.

Among them, Shellac is a natural insect-based material found in a specific tree in the form of resin with unique properties of non-toxicity and edibility.

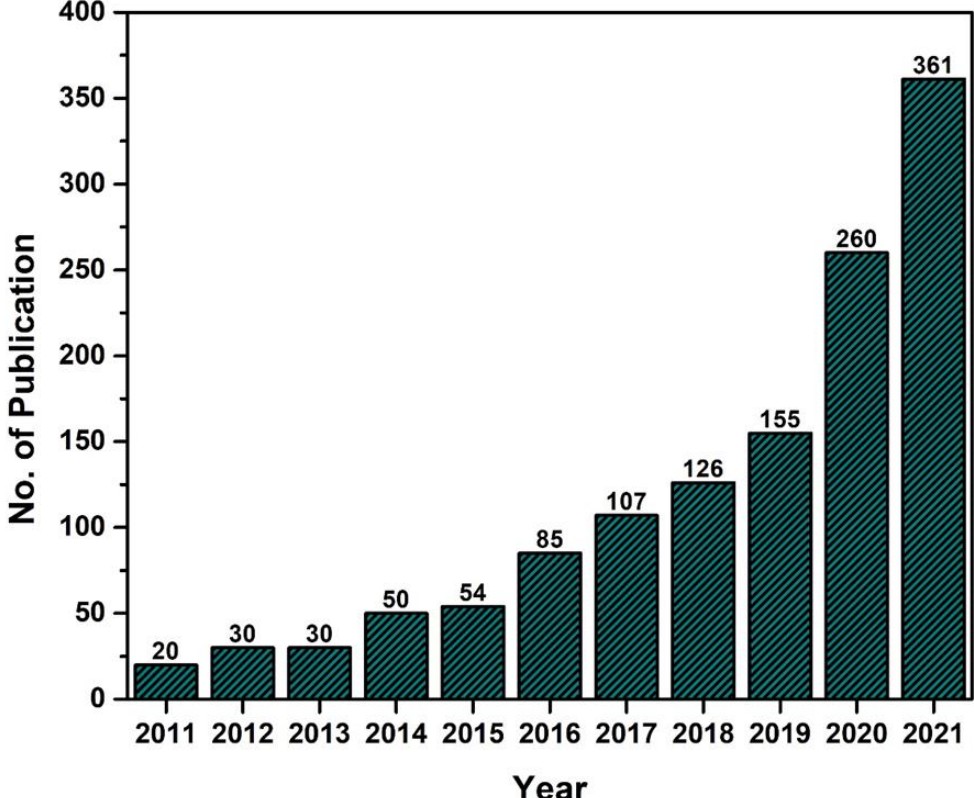

**Figure 1.** Increase in the number of publications on sustainable polymers in packaging. The search was limited to TOPICS in Web of Science, and the keywords used were: biopolymers, bio-based polymers, biodegradable polymer, and packaging.

Shellac, an insect-derived material, has received the least attention due to its scarcity in south Asia [14]. Currently, Shellac is used in various applications, such as furniture polish, glazing agent for candies and pharmaceutical pills, coating on fruits to increase shelf life, primers, smart sensor, 3D printing, and green electronic, as discussed elsewhere [15]. However, the limitations of Shellac include brittleness with time [16], self-esterification [17], low transparency due to orange and brown color [18], solubility in alkaline or acidic medium [19] and in most organic solvents, which have limited its usage in the packaging application. Many of these problems can be improved by physical blending or chemical reaction with other materials to make Shellac more durable, impede self-esterification, and facilitate the film-forming ability, which suggests the potential usage of Shellac in packaging applications.

In this review, we focus on the preparation methods of Shellac, its composition, and its structure. Further, we discuss Shellac's limitations due to its unique chemical structure. Furthermore, the modification of Shellac is discussed, keeping in mind the packaging application of shellac films. The modification of Shellac by using physical blending with plasticizers and other polymers and chemically modifying the shellac structure to improve its properties are discussed. Moreover, using Shellac in other polymers as reinforcement is also studied to showcase its effects on the packaging film and its physical properties. Later, the potential of Shellac to replace the conventional packaging system and its advantages, such as non-toxicity and biodegradability, are considered.

## 2. History of Shellac

Shellac is a hydrophobic, water-insoluble, edible biomaterial with the E number E904, which is used as a glazing ingredient in sweets and pharmaceuticals, as well as a coating substance on fruits to extend shelf life [20]. Shellac's history goes back 5000 years, suggesting it is one of the oldest polymeric resins (Figure 2). According to the Indian mythology "Mahabharata", dried Shellac was mentioned as a material for constructing a palace that was intentionally built to burn down, suggesting the usage of Shellac 5000 years ago [21,22]. Three thousand years ago, the usage of Shellac was evident as "Laksha", and it was used as a source of dyeing agent [23]. In the 11th century, the use of Shellac was mentioned as a source of artist pigment [23,24], whereas Shellac was also utilized as a varnish for cassone [25] and sealing waxes [26] in the 16th century. The highest consumption of Shellac was recorded in the 20th century to produce gramophones [27]. Around 50% of the Shellac was produced to press on the gramophone records in that era.

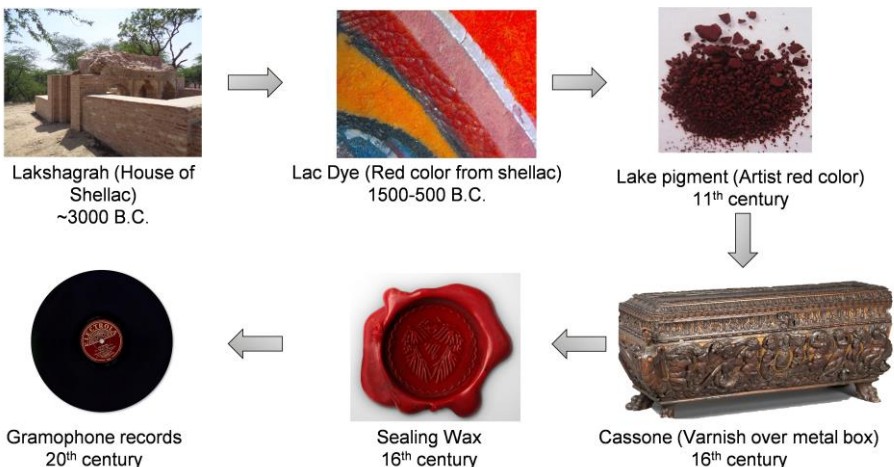

**Figure 2.** History of usage of Shellac and its byproducts [22,28–32].

## 3. Source, Extraction, and Production of Shellac

Shellac is a resin secreted by Kerria Lacca, an insect type of the lac bug [33,34]. Lac means lakh in the Hindi language, which is the root word for Shellac. The word lakh means 100,000 or 0.1 million, which signifies that around 0.1 million lac bugs are required to produce 1 kg of Shellac [15,35]. These lac bugs produce a sticky resin containing Shellac but require various stages to purify it to remove larvae and dyes from it. The intermediate stages of producing Shellac involve the production of resin on the tree (Sticklac) by the lac bug, processing of Sticklac to produce Seedlac, and finally, purifying Seedlac to Shellac.

### 3.1. Production of Resin by Lac Bug on the Tree (Sticklac)

The lac bugs move into specific host trees to consume the sap from the trees to produce the resin. It has also been found that the lac bug, when fed on the Kusum tree (*Schleichera*), produces the least colored Shellac. Other commercial host plants for these lac bugs are Palash (*Butea monosperma*), Ber (*Ziziphus mauritiana*), Kusum (*Schleicher oleosa*), and Semialata (*Flemingia semialata*) [15,36–38]. In the larvae form, the lac bugs move onto the host tree or plant's soft shoots to survive (Figure 3). For 2 to 3 days, they insert their proboscis in the trees to reach the sap. The lac bug then sucks the tree sap to survive and complete its cycle. Meanwhile, the male bug moves out of its cells, and the female bug continues to live in the trees. In order to adhere to the trees for longer, the female lac bug produces a resinous compound as a protective coating around its body. The male bug fertilizes the females and then dies within a few days. The fertilized female continues to secrete the resinous compound and produces 200–500 larvae. This resinous compound is removed by knife, sickle, or stick; therefore, it is termed "Sticklac" [36]. This Sticklac contains resin, wax, dyes, and impurities, such as crusted insect and wood particles.

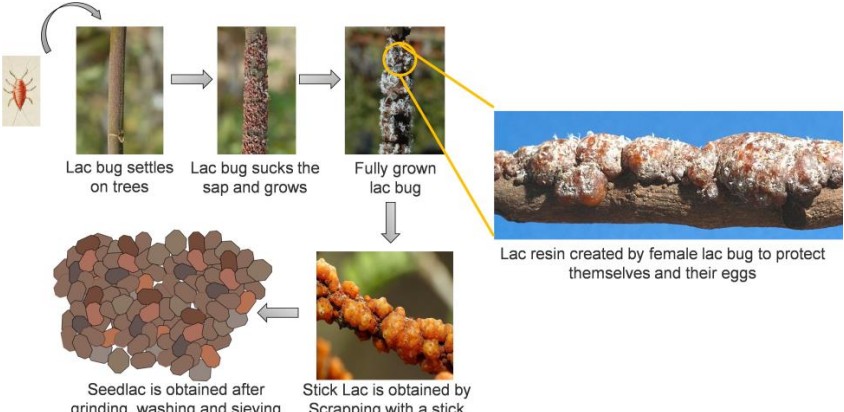

**Figure 3.** Method of production of Seedlac from the lac bug [15,39–41] (reprinted with permission from Thombare et al., Copyright (2022), Elsevier).

### 3.2. Processing of Sticklac to Seedlac

The Sticklac is processed to remove impurities with the help of various processes, such as crushing, washing, drying, cleaning, and grading. The crushing of Sticklac is performed and then sieved with the help of a fine sieve to remove the wood chips. After crushing, the Sticklac is washed with water, and most impurities, such as lac cells, float on the water. The lac that settles at the bottom is called Seedlac, which will be used to produce Shellac [15,36,42].

### 3.3. Production of Shellac by Purification of Seedlac

Seedlac can be transformed into Shellac by two processes, i.e., using thermal and solvent extraction (Figure 4). In the case of the heat process, the Seedlac is placed into a huge cotton bag and heated by charcoal fire to heat it evenly from all sides. The lac resin is melted, squeezed out of the bag, and scraped off the bag's surface. It is then put into hot water to keep the lac molten and distributed into thin sheets. In industries, the Seedlac is melted by heated steam, and molten lac is pressed with the help of the hydraulic press through the filter. This filtered lac is then stretched into long sheets with the help of sheet rollers. In the case of solvent extraction, the Seedlac is dissolved in ethanol, either cold or hot. The soluble resin is dissolved in ethanol, and the insoluble solid is collected from the bottom. The solvents are evaporated, and the remaining viscous Shellac is stretched using a sheeting roller [15,36,42–44]. The solvents are further distilled and can be reused. The sheets obtained through either the thermal process or the solvent process become brittle once dried and form or break in the shape of flakes. Hence, they are sometimes called Shellac flakes.

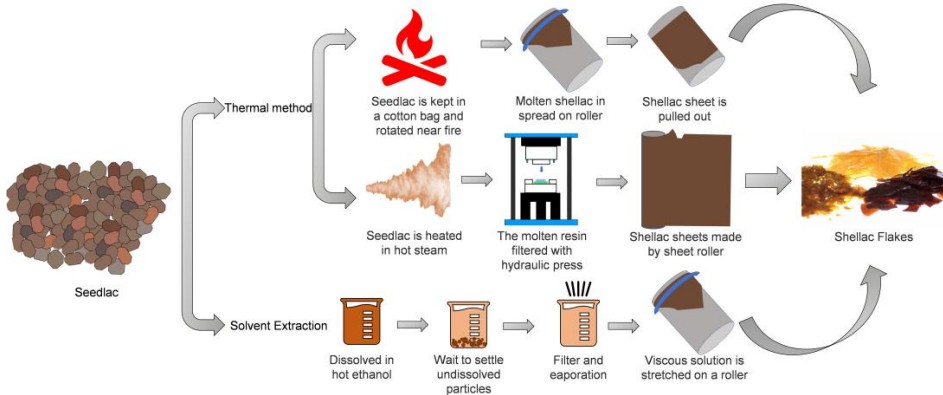

**Figure 4.** Various production methods of Shellac from Seedlac via thermal and solvent methods [45].

## 4. Structural Composition and Structure of Shellac

From Sticklac to Shellac, the main component of Shellac is resin, which is purified in each process. Table 1 shows the different components of Sticklac, Seedlac, and Shellac; for example, the resin in Sticklac is generally 68%, and it is increased to 90.9% in Shellac after processing, while other components, such as dye, wax, and impurities, decrease upon purification.

**Table 1.** Constituents of Sticklac, Seedlac, and Shellac [36] (reprinted with permission from Sharma et al., Copyright (2020), Springer).

| Constituents | Sticklac | Seedlac | Shellac |
|---|---|---|---|
| Resin (%) | 68 | 88.5 | 90.9 |
| Dye (%) | 10 | 2.5 | 0.5 |
| Wax (%) | 6 | 4.5 | 4.0 |
| Gluten (%) | 5.5 | 2 | 2.8 |
| Foreign bodies (%) | 6.5 | - | - |
| Impurities (%) | 4 | 2.5 | 1.8 |

The chemistry of the Shellac resin is very complex, yet captivating. The chemistry of resin has been studied since the 1960s by various techniques, such as column chromatography, thin-layer chromatography, gas chromatography, and spectroscopic methods. Shellac is a complex polyester resin composed of long-chain hydroxy-fatty and sesquiterpene acids. It contains inter- and intra-esters of polyhydroxy carboxylic acids. Typically, the resin has five free hydroxyls, one free carboxylic, and one aldehyde, partly free and partially combined. It also has linkages of esters, acylal, acetal, and ether. One-third of the hydroxyl groups in Shellac resin are free; the rest are combined and identified with the reaction with periodic acid [36]. The backbone of Shellac comprises long-chain fatty hydroxy acids known as Aleuritic acids, which comprise 35% of Shellac and impart hydrophobicity. This backbone is connected with many cyclic terpene acids (hydrophilic) with ester bonds. The terpemnic acids identified in the Shellac resin were jalaric, shellolic, laksholic, laccijalaric, laccishellolic, and laccilaksholic acids [46]. The combination of these acids makes Shellac amphiphilic and provides distinctive properties for a wide variety of applications [15]. Although most of the constituents of Shellac resin are known, its structure is entirely unknown [36]. However, some researchers have tried to conclude that in pure Shellac resin, aleuritic acid and terpenic acids are present in a 1:1 ratio, where there are three jalaric/epishellolic acids and one laccijalaric/epilaccishollolic acid [36,47]. The chemical structure of Shellac concluded as the most possible structure is shown in Figure 5.

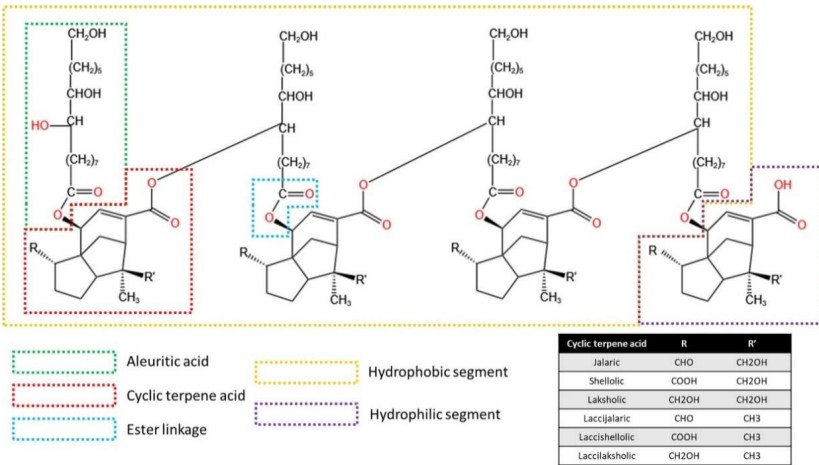

**Figure 5.** Chemical structure of Shellac.

## 5. Physicochemical Properties of Shellac

Shellac has unique properties due to its complex structure. The various properties of Shellac are summarized in Table 2. Shellac is commercially obtained in the form of flakes. It is generally yellow to brown, depending on the dyes, which are determined by the tree. It has a refracted index in the range of 1.521 to 1.527, near that of glass. Due to the presence of acids in its chemical structure, it is acidic and has a high acid value ranging from 60 to 75. Upon storage, the acid value decreases due to the reaction between its free hydroxyl groups and carboxylic acid [48]. This phenomenon of Shellac is called self-esterification or self-polymerization, due to which Shellac becomes hardened as it ages; therefore, the high quality of Shellac can be estimated with the help of its acidic value [15,49]. On the other hand, the saponification value of Shellac is reported as 225–230, indicating that it has lower fatty acids and molecular weight. The molecular weight of Shellac varies depending upon the type of tree used by the lac bug. The molecular weight of hard resin in Shellac is 2000–2210, and that of soft resin is 500–550. The hard resin makes up nearly 70% of Shellac, whereas the rest is soft resin. The average molecular weight of Shellac is 1000–1006, which indicates that Shellac falls in the resin category. The basicity of Shellac is 2, denoting that Shellac is dibasic and can donate two protons. The ester, hydroxyl, and carboxyl values are in the range of 155–165, 250–280, and 7.8–27.5, respectively [15]. It is generally soluble in alcohols and organic solvents and shows film-forming ability when dissolved in ethanol or alkaline solutions [15,50–52]. It is, however, insoluble in hydrocarbon solvents, esters, and water. Moreover, the insolubility of Shellac in water has led to the creation of nanoparticles by antisolvent precipitation methods using ethanol and water systems in the presence of gums [53].

**Table 2.** Physicochemical properties of Shellac.

| Property | Description | Reference |
|---|---|---|
| Appearance | Flakes | [54] |
| Type | Complex polyester | [15] |
| Color | Yellow to brown | [23,54] |
| Refractive index | 1.521–1.527 | [15] |
| Acid value | 60–75 | [15,50] |
| Saponification | 225–230 | [50] |
| Ester value | 155–165 | [15] |
| Hydroxyl number | 250–280 | [15] |
| Carboxyl value | 7.8–27.5 | [15] |
| Basicity | 2 | [15] |
| Molecular weight | 1000–1006 | [15] |
| Solubility | Insoluble in water, hydrocarbon solvents, and esters Soluble in alcohol and organic solvent | [15,49] |
| Density | 1.035–1.21 | [15] |
| Tensile strength | 5.7–14 MPa | [15,55] |
| Young's modulus | 338.4 MPa | [56] |
| Elongation at break | 3.05 % | [56] |
| Stress | 2.25–2.5 MPa | [52,57] |
| Strain | 3.05–3.5% | [52,57] |
| Puncture strength | 3.8 MPa | [58] |
| Puncture elongation | 4% | [58] |
| WVPC | $(4–5.5) \times 10–9$ gm/h-mm-mmHg | [52,58,59] |
| Glass transition | 38–40 °C | [15] |
| Softening point | 65–70 °C | [15] |
| Melting | 75–90 °C | [15] |
| Aging/cross-linking | 45 min at 150 °C | [60] |
| Decomposition temperature | 280 °C (Onset TGA) | [61] |

The density of Shellac ranges from 1.035 to 1.20 gm/cm$^3$, which makes it sink in water, thus limiting water pollution, unlike most polymers used in packaging, such as polyethylene (PE) and polypropylene (PP) [62] with density lower than 1 gm/cm$^3$. Shellac's tensile strength ranges from 5.70 to 14 MPa, near the value of LDPE and PHB [63,64]. Shellac's stress and strain values are 2.25–2.50 MPa and 3.05–3.50 %, respectively. Shellac's puncture strength and elongation are near 3.8 MPa and 4%, respectively. The mechanical property of Shellac shows that it is brittle and less elastic. The thermal property, such as the glass transition temperature of Shellac, is in the range of 38–40 °C, making it brittle at room temperature and soft at 65–70 °C, near its melting point of 75–85 °C. According to the onset temperature in TGA, the decomposition temperature is 280 °C, near that of PHB [65,66]. With aging, Shellac becomes stiff, brittle, and insoluble in solvents due to the inter-esterification of the free hydroxyl groups with the free acids groups [67], which limits its consumption for several applications. It was also reported that higher temperatures accelerate the aging of Shellac. For example, heating Shellac at 150 °C for 45 min or at 175 °C for 15 min completely ages Shellac [60]. This aging is also called the polymerization of Shellac, and the molecular weight increases by many folds [15]. During the aging process, the Shellac is melted from the solid state and flows readily, but with time, the molten Shellac forms a rubbery state, an intermediate stage of aged Shellac. After some time, the rubbery state converts to a hard and brittle structure. This aged Shellac is also called polymerized Shellac due to the self-esterification, as shown in Figure 6. To determine the percentage of aging or cross-linking, the aged Shellac is dissolved in ethanol, and the amount of insoluble content shows the amount of Shellac that has been aged. Therefore, the usage of Shellac is limited by its storage condition.

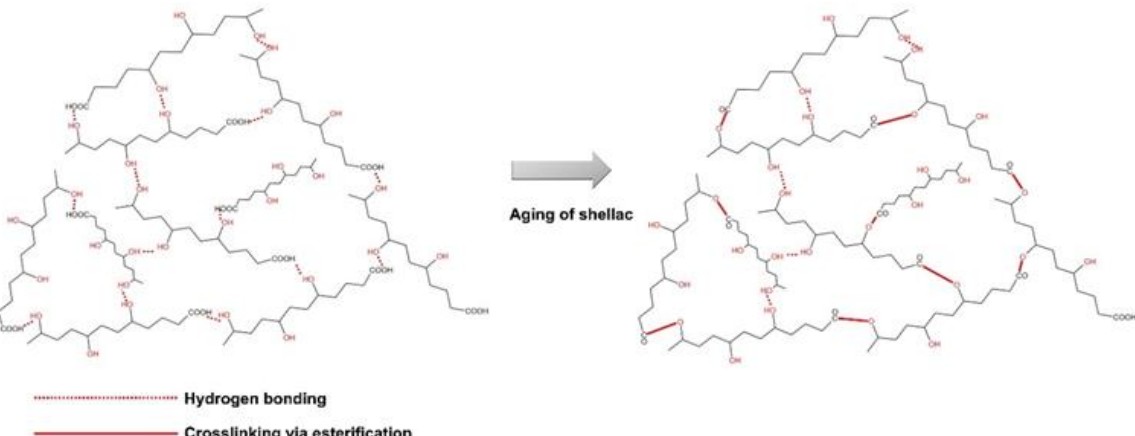

**Figure 6.** Aging of Shellac by cross-linking or inter-esterification.

## 6. Modification of Shellac

In packaging, Shellac is used either as a direct coating on fruits or for film formation. The usage of Shellac in the food industry is discussed elsewhere [46]. Shellac is a resin, but it also has a unique film-forming ability. When dissolved in an ethanolic or alkaline solution, Shellac can be readily solvent cast to produce Shellac films. The major problem in the solvent casting of shellac film is that it adheres to the Petri dishes due to its inherent tackiness. Therefore, the film formation is performed on Teflon-coated or silicon-coated plates, so the film can be removed easily [55,58,59]. Shellac produces different properties in alkaline solutions than ethanol due to the salts in alkaline solutions [57]. These salts interact with the functional groups of Shellac and thus produce different properties. The salt solution, such as ammonium salt, when used to develop the films, is not quickly aged, as the ammonium salt protects the carboxyl site of Shellac and thus retards the polymerization. However, the Shellac films cast using an ethanolic solution resulted in a higher water barrier than those cast using an alkaline solution [57]. An alkaline solution

makes Shellac soluble in water, and it loses its water repellency. Therefore, solvent usage can affect the film-forming ability and properties of the films. To make Shellac compatible with packaging applications, it should have good barrier and mechanical properties, and the film should not age by self-crosslinking. Therefore, Shellac has been modified in the past by many methods to make it compatible with the packaging application.

*6.1. Physical Mixing*

6.1.1. Addition of Plasticizers

Shellac was mixed with several plasticizers to increase the flexibility and thereby retard the self-polymerization of the films with time, as mentioned in Table 3. Triacetin (TA) [57], diethyl phthalate (DEP) [57], triethyl citrate (TEC) [56], and polyethylene glycol (PEG) [52,56] were used as plasticizers for Shellac films in the past. The plasticized Shellac films showed lower stress and elastic modulus values and increased strain and elongation at break. TA, DEP, and TEC, when used at the concentration of 10 wt.% of Shellac in films, were able to plasticize Shellac easily with a maximum strain of 62% in the case of TA. TEC was the least effective with a similar strain, but when used at 30 wt.%, the strain of plasticized Shellac was increased to 22.35%. Plasticizers are known to increase the water vapor permeability coefficient (WVPC) [68], which was easily seen in the case of TEC. However, interestingly, the water vapor permeability coefficient was slightly reduced with the addition of TA and DEP, which was due to the lower molecular weight of the TA and DEP plasticizer. Moreover, DEP addition did not decrease the aging of Shellac as significantly as TA and PEG (Figure 7d); this was due to the poor interaction and low affinity of DEP with the Shellac network, which were visible in the FE-SEM (Figure 7a–c) [57].

**Table 3.** Effect of plasticizers on the Shellac films (N.A.: Data Not Available).

| Plasticizer | wt.% of Shellac | Stress (MPa) | Strain (%) | Elastic Modulus (MPa) | Elongation at Break (%) | WVPC × $10^9$ (g/h-mm-mmHg) | Reference |
|---|---|---|---|---|---|---|---|
| Triacetin | 10 | 1.60 | 62 | N.A. | N.A. | 4.5 | [52] |
| Diethyl Phthalate | 10 | 1.75 | 58 | N.A. | N.A. | 4.8 | [52] |
| Triethyl citrate | 10 | N.A. | N.A. | 291 | 3.1 | N.A. | [56] |
| Triethyl citrate | 30 | N.A. | N.A. | 108.6 | 22.35 | N.A. | [56] |
| PEG 200 | 10 | 0.25 | 98 | N.A. | N.A. | 3.25 | [57] |
| PEG 400 | 10 | 1.6–1.75 | 40–45 | N.A. | N.A. | 6.2–6.5 | [52,57] |
| PEG 600 | 10 | N.A. | N.A. | 308.8 | 3 | N.A. | [56] |
| PEG 600 | 30 | N.A. | N.A. | 40.1 | 85 | N.A. | [56] |
| PEG 1500 | 10 | N.A. | N.A. | 338.4 | 3.1 | N.A. | [56] |
| PEG 1500 | 30 | N.A. | N.A. | 37.9 | 117.3 | N.A. | [56] |
| PEG 4000 | 10 | 0.25 | 22 | 443.7 | 2.9 | 7 | [56,57] |
| PEG 4000 | 30 | N.A. | N.A. | 25.7 | 153 | N.A. | [56] |
| PEG 6000 | 10 | N.A. | N.A. | 495 | 2.95 | N.A. | [56] |
| PEG 6000 | 30 | N.A. | N.A. | 24.5 | 160 | N.A. | [56] |

A study showed that PEG 200 and PEG 400 had dramatically affected the strain and increased the strain to 98% and 45%, respectively. However, the elongation at break of plasticized Shellac was not affected by the PEG 600–35,000 when used at 10 wt.% of shellac. At higher concentrations, i.e., 30 wt.%, the PEG 600 to PEG 35,000 increased the strain of plasticized Shellac to a maximum of 179%. With the addition of PEG, the WVPC increased with an increase in PEG's molecular weight. This increase in strain and decrease in the water barrier were in line with the increase in molecular weight of the PEG used. When comparing PEG 200, PEG 400, and PEG 4000 at 10 wt.% of Shellac, PEG 400 was able to reduce the aging of Shellac compared to PEG 4000 and PEG 200. PEG 200 might not have a sufficient chain length to disrupt the self-esterification of Shellac, and PEG 4000's longer chain length could not have infiltrated the shellac network; hence, PEG 400 modified shellac's solubility in ethanol was the highest after the aging test representing lesser crosslinking (Figure 7e) [52].

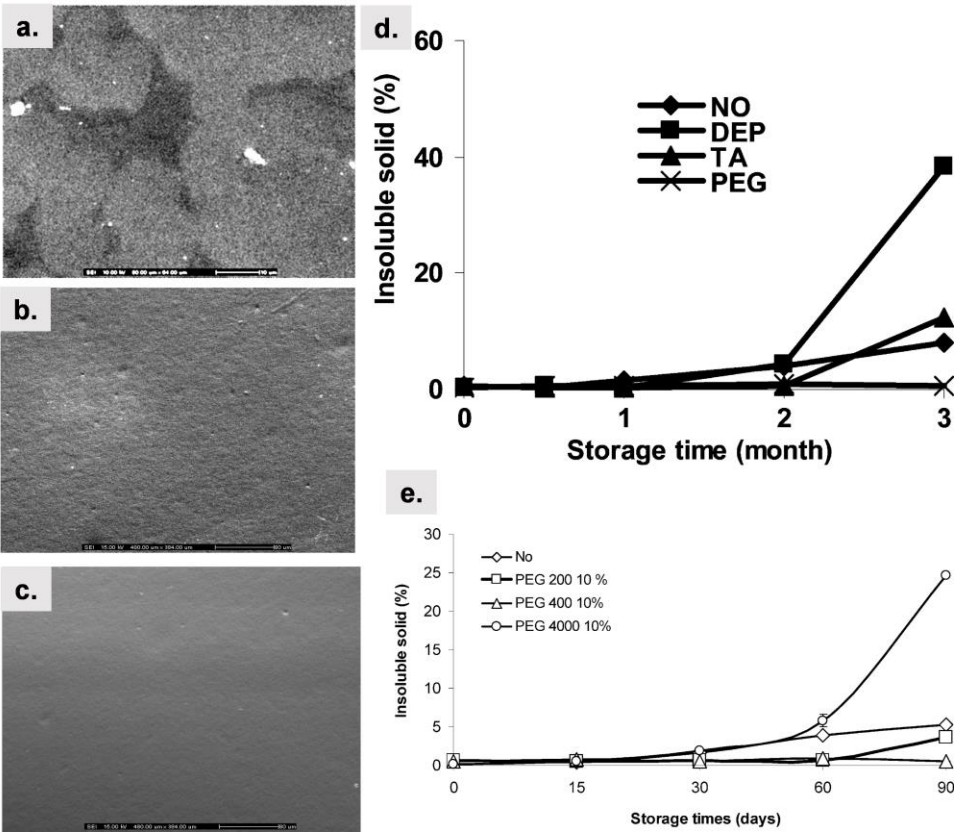

**Figure 7.** FE-SEM images of shellac matrix with plasticizer (**a**) DEP, (**b**) TA, and (**c**) PEG 400, (**d**) Insoluble content representing aging of Shellac with DEP, TA, and PEG [57] (reprinted with permission from, Luangtana-anan et al., Copyright (2007) American Chemical Society); (**e**) Insoluble content representing aging of Shellac with PEG 200, PEG 400, and PEG 4000 [52] (reprinted with permission from, Luangtana-anan et al., Copyright (2010) American Chemical Society).

6.1.2. Blending with Polymers

Adding different polymers to the Shellac has also been used to alter the properties mentioned in Table 4. Cellulose derivatives, such as hydroxypropyl methylcellulose (HPMC), methylcellulose (MC), and ethylcellulose (EC), were used as reinforcement to alter the properties of Shellac. HPMC and MC, when used at 20 wt.%, increased the elongation at break to 22.3% and 140.3%, respectively. MC, which had caused the highest plasticization, decreased the Tg to 35 °C [56]. Both of these cellulose derivatives were able to plasticize the Shellac, as it was hydrophilic, unlike the EC, which increased the elastic modulus and lowered the elongation at break. Another study [59], where EC was employed with varying weight percentages of Shellac, shows that an increase in EC decreases the strain and increases the stress. However, due to EC's inherent hydrophobic nature, the WVPC of the composite film was slightly decreased. Carbopol 940 (carbomer), a polymer of acrylic acid, increased the elongation at break up to 31.91%, which was higher when compared to HPMC but lower than MC when added to the Shellac film. It was worth noting that only 2 wt.% of carbomer increased the elongation at break of the Shellac films. Meanwhile, the PVA was not able to increase elasticity and was thus unable to plasticize the Shellac. Being water soluble, PVA could not plasticize the Shellac, since its own Tg is near 81 °C [69]; thus, its addition increased the Tg of Shellac films to 72 °C, making it brittle [56]. Gelatin, a protein used to plasticize the PLA [70], also plasticizes the Shellac film. The puncture elongation increased from 3.8% to 26.66% when gelatin was added to Shellac at 40 wt.%. However, the puncture strength also increased, showing the effect of amino groups of gelatins interacting with the hydroxyl and carbonyl groups of Shellac [58]. However, the WVPC of the shellac–gelatin film increased, showing a decrease in the water barrier due

to the high polarity of the gelatin. PEG 400 and DEP were added to the Shellac–gelatin film to study the effect of the plasticizer. It was found that the addition of PEG 400 and DEP increased puncture elongation further to 134.28% and 108.17%, respectively, while decreasing the puncture strength due to the drop of electrostatic interaction in the polymeric chains. This was due to the interference of the plasticizers in the interaction of gelatin and Shellac. An interesting phenomenon of DEP was noted in the Shellac–gelatin film, namely that it acted like a plasticizer by increasing the elongation and improving the WVPC of the composite films compared to PEG 400. The WVPC of the PEG 400 plasticizer increased by 29.9%, but the WVPC decreased by 64.02% with DEP. This increase in the water barrier was due to the hydrophobic nature of DEP.

**Table 4.** Effect of polymer on the Shellac films (N.A.: Data Not Available).

| Polymer | wt.% of Shellac | Tensile Strength (MPa) | Elongation at Break (%) | Elastic Modulus (MPa) | Puncture Elongation (%) | Puncture Strength (Mpa) | WVPC × 10⁹ (g/h-mm-mmHg) | Reference |
|---|---|---|---|---|---|---|---|---|
| HPMC | 20 | N.A. | 22.3 | 144 | N.A. | N.A. | N.A. | [56] |
| MC | 20 | N.A. | 140.3 | 63.1 | N.A. | N.A. | N.A. | [56] |
| EC | 20 | N.A. | 2.8 | 360.1 | N.A. | N.A. | N.A. | [56] |
| EC | 40 | 12.28 | 1.67 | N.A. | N.A. | N.A. | 4.09 | [59] |
| Carbomer | 2 | N.A. | 31.91 | 110.2 | N.A. | N.A. | N.A. | [56] |
| PVA | 20 | N.A. | 2.8 | 489 | N.A. | N.A. | N.A. | [56] |
| Gelatin | 40 | N.A. | N.A. | N.A. | 26.66 | 14.07 | 6.12 | [58] |
| Gelatin+ PEG 400 (10%) | 40 | N.A. | N.A. | N.A. | 134.28 | 3.71 | 7.95 | [58] |
| Gelatin + DEP (10%) | 40 | N.A. | N.A. | N.A. | 108.17 | 7.34 | 2.86 | [58] |

*6.2. Chemical Modification*

In the past, chemical modifications of polymers have been implemented to enhance the physicochemical properties of the polymers. Shellac has also been modified by various methods, such as electron beam, UV radiation, grafting, esterification, and many more, as mentioned in Table 5. Various methods can easily modify Shellac, as it has many different functional groups, such as hydroxyl and carboxylic acids. Grafting of 2-hydroxyethyl methacrylate (HEM) by ultraviolet (UV) radiation was performed on Shellac (Figure 8a). The acrylic group of HEM was grafted onto the hydroxyl group of Shellac to change its physicochemical properties in the presence of UV radiation at varying times. The double bond of HEM was broken, and then, the acrylic group had grafted onto the hydroxyl group of Shellac [55]; thus, the formation of ether bonds confirmed the reaction, and the tensile strength increased by 36.8%, and elongation at break also increased by 17.64%. This grafting also reduced the aging process of Shellac, as the self-polymerization of Shellac did not occur due to the consumption of free hydroxyl bonds [55].

**Table 5.** Chemical modifications of Shellac films (N.A.: Data Not Available).

| Modifier | Weight (%) | In Presence of | Reaction | Chemical Change | Tensile Strength (MPa) | Elongation at Break (%) | Puncture Strength (MPa) | Puncture Elongation (%) | Reference |
|---|---|---|---|---|---|---|---|---|---|
| HEM | 3 | UV Irradiation | 6 h | Grafting with acrylic (Ether bonds) | 7.8 | 2 | N.A. | N.A. | [55] |
| HEM | 5 | Gamma Radiation | 1 kGy | Grafting with acrylic (Ether bonds) | 8.59 | 4.36 | N.A. | N.A. | [71] |
| EHA | 5 | Gamma Radiation | 1 kGy | Grafting with acrylic (Ether bonds) | 9.31 | 3.356 | N.A. | N.A. | [71] |
| BDDA | 5 | Gamma Radiation | 1 kGy | Grafting with acrylic (Ether bonds) | 11.96 | 7.28 | N.A. | N.A. | [71] |
| Succinic anhydride | 72.86 | 60 °C | 6 h | Ester | N.A. | N.A. | 6 | 88 | [72] |
| Jeffamine D-2000 | 20 | 100 °C | 4 h | amine-carboxyl reaction | N.A. | N.A. | 3.5 | 85 | [51] |

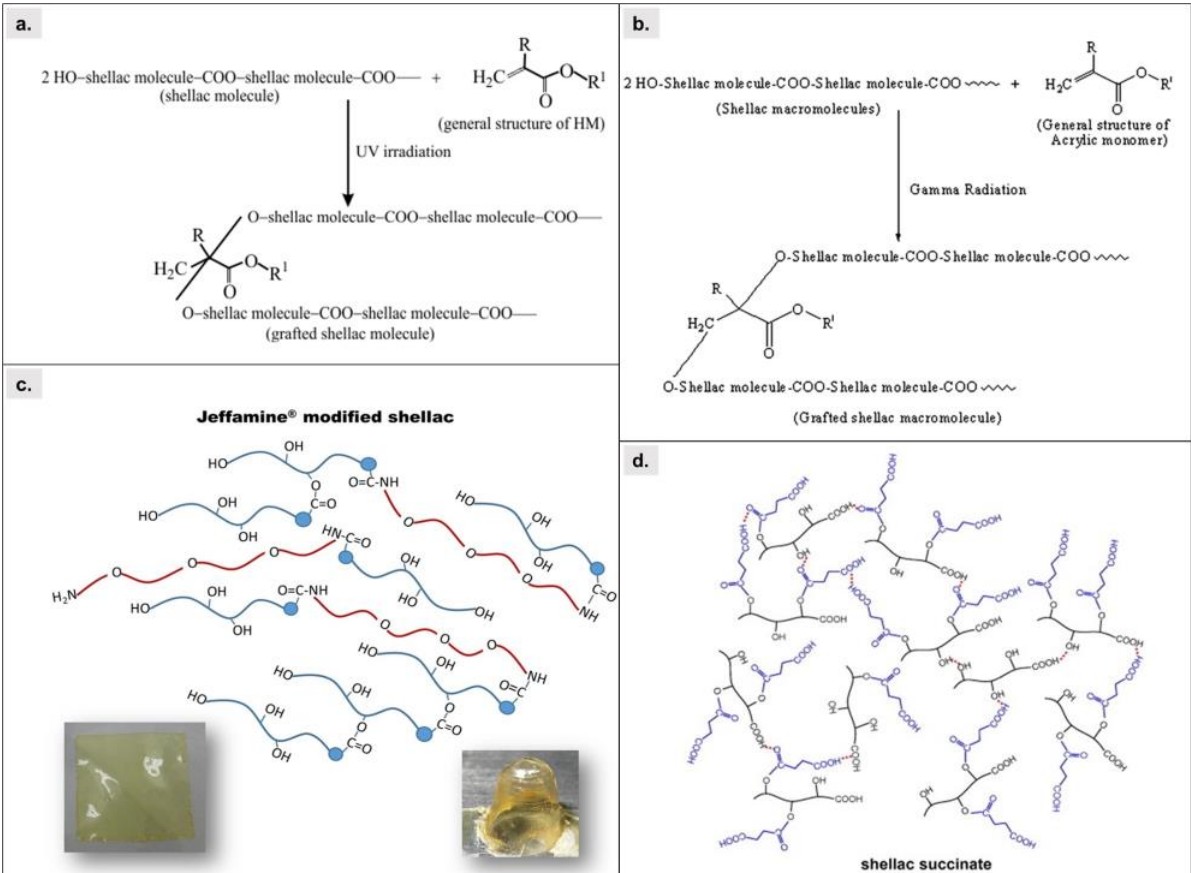

**Figure 8.** Chemical modification of Shellac with (**a**) HEM via UV radiation [55] (reprinted with permission from Arnautov et al., Copyright (2013) Springer); (**b**) Acrylic monomer via Gamma radiation [71] (reprinted with permission from, Ghoshal et al., Copyright (2010) Springer); (**c**) Jeffamine (formation of C=O-NH) [51] (reprinted with permission from, Bar et al., Copyright (2010) Springer); and (**d**) Succinic anhydride (formation of C=O-OR) [72] (reprinted with permission from, Limmatvapiratal et al., Copyright (2008) Springer).

Similarly, HEM, 2-Ethylhexyl acrylate (EHA) and 1, 4 butanediol diacrylate (BDDA) were grafted on the Shellac via gamma radiation at a dose rate of 3.5 kGy/hr [71]. The mechanism was similar to that of HEM, i.e., the formation of ether bonds by the reaction of acrylic group onto the hydroxyl groups of the Shellac (Figure 8b). The highest increase in tensile strength and elongation at break was observed with the grafting of BDDA due to the presence of two acrylic functional groups in BDDA compared to EHA and HEM, which has only one acrylic functional group. Succinic anhydride was also grafted onto Shellac to consume its hydroxyl bonds in an esterification reaction, which led to increased puncture strength and puncture elongation of nearly 6 MPa and 88%. The higher elongation of shellac succinate might be due to the succinate moieties formed by the ester linkages of succinate acid and hydroxyl groups of Shellac (Figure 8d). However, the WVPC increased by 16.18%, showing a poor barrier, which might be due to the free carboxyl acids of Succinic anhydride, which did not participate in the reaction, as it has two carboxyl groups at its ends. Therefore, the solubility of grafted Shellac at pH 7 also increased, showing high polarity [72]. In another study, Jeffamine modifiers were used to produce flexible Shellac. Jeffamine are a group of polymers or modifiers with amine groups with the backbone of polyether. The amine group of Jeffamine, when reacted with the carboxyl groups of Shellac, led to the formation of amide bonds (Figure 8c); hence, the free carboxyl groups were lower and were thus able to retard the inter-polymerization of the Shellac and acted as a plasticizer [51]. Jeffamine-D2000, which has 2-amine functionality and has propylene oxide

backbone, was used to modify the raw Shellac at 100 °C. It produced a flexible film with a high puncture elongation of nearly 85% and a puncture tensile strength of 3.5 MPa.

## 7. Shellac as Reinforcement in Other Polymers

Shellac has many different functional groups; thus, it has been utilized as a filler or as reinforcement in other bio-based polymers to improve their physicochemical properties, especially the water vapor barrier, as mentioned in Table 6. In the case of HPMC, Shellac had increased the water vapor barrier by 23% just by incorporating 0.5 wt.% of the total HPMC matrix. A small amount of Shellac improved the HPMC polymer's barrier with the help of lauric acid, which was used as an emulsifier (0.025 wt.%) for better distribution. However, Shellac decreased the elongation and tensile strength by 37% and 23%, respectively, due to the incompatibility [73]. However, in the case of konjac glucomannan (KGM) polymer film, the tensile strength and elongation increased by 253% and 100%, respectively. Researchers suggested that the increase in tensile strength might be due to the cross-linking of Shellac with KGM, and the elongation might have increased due to Shellac's low molecular weight, so it acted as a plasticizer in the polymer matrix [74].

**Table 6.** Effect of Shellac on the physical properties of different polymers (N.A.: Data Not Available).

| Matrix | Shellac | Other Additives | Elongation | Moisture Barrier | Tensile Strength | Stress | Strain | Reference |
|---|---|---|---|---|---|---|---|---|
| HPMC | 0.5% | 0.025% Lauric Acid | Decreased by 37% | Increased by 1.36 times | Decreased by 23% | N.A. | N.A. | [73] |
| KGM | 1 gm | 400 µL glycerol | Increased by 100% | Increased by 1.28 times | Increased 253% | N.A. | N.A. | [74] |
| Chitosan | Nano shellac | 300 µL glycerol | Decreased by 36% | Increased by 1.4 times | Increased by 40% | N.A. | N.A. | [75] |
| Pectin | 30% | N.A. | N.A. | Increased by 2.19 times | N.A. | Decreased by 60% | Decreased by 66.6% | [76] |
| Pea starch/guar gum/glycerol | 40% | 1% Stearic acid, 0.3% Tween-20 | Increased by 31.12% | Increased by 68 times | Decreased by 40.43% | N.A. | N.A. | [77] |
| Casein | 10% | N.A. | N.A. | Increased by 1.33 times | Decreased by 42.7% | N.A. | N.A. | [78] |
| Soybean protein isolate | 9.6 wt.% | 30 wt.% Glycerol | Decreased by 28.80% | Increased by 2 times | Increased by 17.58% | N.A. | N.A. | [79] |
| CMC | 20% | N.A. | Deceased by 19.37% | N.A. | N.A. | N.A. | N.A. | [80] |

To create an entirely edible biopolymer film, pectin was modified by Shellac. By adding Shellac, a decrease in stress and strain was observed of 66.6% and 66%, respectively. The strain decreased due to Shellac's inherent brittle nature; however, the stress of the pectin/shellac composite was also decreased due to lower hydrogen bonding, as Shellac was less polar than pectin. However, the lower polarity of Shellac decreased the polarity of the film from 46% to 32.68%, which increased the water barrier by 54.5% [76]. Apart from pectin, pea starch with guar gum was fabricated with Shellac for use in food packaging. In the case of pea starch/guar gum, the tensile strength decreased, and the elongation at break increased, which was opposite to pectin and HPMC, as discussed above. This anomaly with pea starch and guar gum was due to the role of glycerol, which acted as a plasticizer, and the presence of stearic acid, a type of long-chain fatty acid, which acted as an emulsifier for better film formation. The presence of stearic acid produced a rough surface due to heterogeneous lipid distribution within the film [77], as seen in the FE-SEM images (Figure 9a). However, it was also worth noting that the addition of stearic acid increased the barrier of the pea starch/guar gum/shellac films by 1.65 times, and Shellac alone increased the barrier by 42 times; therefore, the water barrier increased by 68 times [77]. When carboxymethyl cellulose (CMC) was mixed with Shellac, the pores of CMC were filled with Shellac, as seen in FE-SEM (Figure 9b), which was responsible for a high barrier and elongation decrease by 19.37%. With Shellac and CMC at a 1:1 ratio, the film formed was brittle and had higher burst properties. It was suggested that CMC would have interacted with the side chain groups of Shellac for enhanced elongation [80]. Casein—a type of protein and another edible polymer but structurally different than the previously discussed starch and pectin, which were polysaccharides—was modified with Shellac. In the casein/shellac composite film, eugenol was added for antimicrobial activity. The tensile strength of casein/shellac decreased compared to casein alone due to the shielding of calcium ions by the lower molecular weight of Shellac. The tensile strength

reduced further with increasing shellac content, confirming the interaction of Shellac with casein. As expected, the water barrier increased 1.33 times due to decreased polarity [78]. Another type of protein, soy protein isolate (SPI), increased the water barrier by two times when mixed with Shellac. Infrared spectroscopy explained that the amine group of SPI was replaced, and hydrogen bonding was formed. The increased hydrogen bonding was responsible for increased barrier and tensile strength. The tensile strength was increased by 17.58%, and elongation was reduced by 28.8%. The composite films showed lower water absorption and higher contact angle due to the addition of Shellac [79]. Apart from the direct blending of Shellac in another polymer matrix, nano-shellac was created using the precipitation method and incorporated into the chitosan polymer matrix. The incorporation did not affect the surface morphology much, and the matrix had compact morphology (Figure 9c). The shellac nanoparticle increased the tensile strength, as the nanoparticles strengthened the chitosan matrix, and the elongation at break was reduced. The water barrier eventually increased due to the increase in compactness of the structure [75].

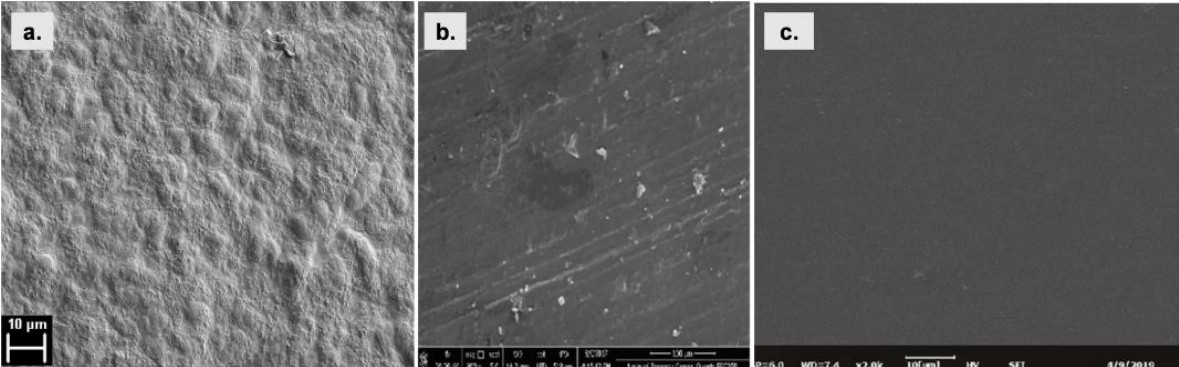

**Figure 9.** Surface morphology study through FE-SEM micrographs of Shellac reinforced in (**a**) Pea starch/guar gum/glycerol [77] (reprinted with permission from, Saberi et al., Copyright (2017) Elsevier); (**b**) Shellac reinforced in CMC matrix [80] (reprinted with permission from, Mohamed et al., Copyright (2019) Elsevier); (**c**) Shellac nanoparticles in chitosan matrix [75] (reprinted with permission from, Yuan et al., Copyright (2021) Elsevier).

## 8. Toxicity, Biodegradability, and Compostability

Shellac is a food-grade substance that is non-toxic and has been proven safe for people, so it has been used to coat pharmaceutical tablets, sweets, and fruits. Toxicity is a crucial concern regarding edibility; nevertheless, Shellac was proven non-toxic upon evaluation in chronic toxicity research involving rats fed over 180 days [81]. The clinical symptoms, feed intake, body weight increase, organ body weight ratio of rats, necropsy, and histological testing revealed no abnormalities. It was determined that Shellac feed concentrations up to 5000 ppm are safe and do not cause toxic manifestations. In another study, a substance called "Shellac F" (Sodium fluoride (5%), Shellac, modified epoxy resin, acetone, and silica) was examined for cytotoxicity to see whether it could be used as a desensitizing agent for dentin hypersensitivity [82]. It was shown to be less harmful than the commercial desensitizing agent "Isodan". In another study, Shellac micro-hierarchical films were examined for drug release and found safe for clinical trials. Shellac has been employed as a coating material for enteric characteristics due to its non-toxicity and solubility in alkaline pH. Shellac is solubilized in our intestine, which is alkaline, rather than in our stomach, which produces acid.

Apart from being non-toxic, Shellac has been mentioned as biodegradable by some reviews [15,46] and past researchers' texts [83–86]. The biodegradation research on Shellac was conducted in a study in which the Shellac was grafted with cyclic monomers using gamma radiation. This study evaluated and modified Shellac for biodegradability in soil for 30 days. It was observed that Shellac lost 11.3% of its weight, whereas modified Shellac lost 68.7% of its weight. As a result of grafting with acrylic, the biodegradation rate of

Shellac was enhanced [71]. The non-toxicity and biodegradability of Shellac make it a better available natural and renewable source for packaging material.

## 9. Potential in the Packaging Application

Shellac is a resin primarily used in varnishes nowadays, but its applications are not limited. The benefit of Shellac as a primer, binder, coating, glaze enhancer, adhesive, cosmetics, food, pharmaceutical products, textiles, adhesives, plastic, rubber, leather, fertilizers, seeds, fruits, wood, pyrotechnics, printer inks, paints, and confectionery depends on the properties and grade of Shellac, which is best suited for the specific products. Aside from these applications, Shellac has been researched for packaging applications, but this has been limited to 20 publications, as shown in Figure 10. Those 20 publications account for 1.56% of the articles covering bio-based polymers for packaging applications. However, although the publications are few in number, they are increasing year by year.

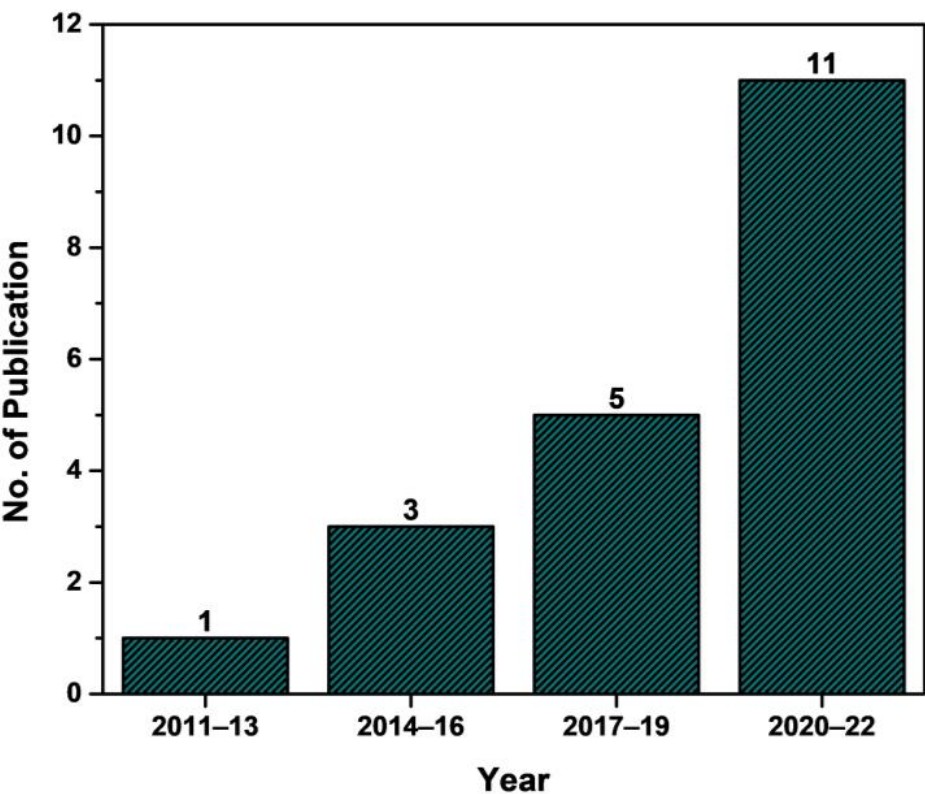

**Figure 10.** Increase in number of publications on Shellac usage in packaging. The search was limited to TOPICS in Web of Science, and the keywords used were Shellac and Packaging.

Recently, the literature on Shellac for packaging applications has included a study on pulp-based cutlery, a coating solution for active packaging, and as reinforcement in other bio-based polymer matrices. For water resistance, Shellac was coated on pulp-based cutlery; however, the literature indicates mixed results. In one research work, Shellac failed to serve as a water barrier in some way [87]. In contrast, in another study, Shellac with nano-fibrillated cellulose (NFC) was coated on pulp cutlery, improving the water barrier and outperforming traditional polymers, such as polyethylene terephthalate (PET) [88]. With various biopolymers, including pea starch/guar gum [77], KGM [74], CMC [80], and SPI [79], Shellac has been used as reinforcement to strengthen the water barrier, oxygen barrier, and mechanical properties, which are crucial for the majority of food packaging applications. Shellac, a useful reinforcement material for food packaging, also boosted the temperature tolerance of the packaging film [74]. To retain the integrity of the packaged product, Shellac has been researched in active packaging alongside olive leaf/grape pomace extract [89] and pine needle essential oil [90] as a coating solution. Eggs were coated with

Shellac and pine needle essential oil to preserve the eggs, improve the ultraviolet resistance, and increase the water and oxygen barrier performance [90]. Additionally, eugenol was added to PVP/shellac fiber films for the packaging of strawberries [91]. Moreover, to extend the shelf life of fruits, 1-Methylcyclopropene was applied on paper as a coating material using Shellac as a base matrix [92].

Shellac's potential among traditional crude-based polymers and conventional bio-based polymers must be demonstrated by properties critical to packaging, such as melting temperature, tensile stress, elongation at break, contact angle, and WVPC. Table 7 compares Shellac to different polymers and indicates that Shellac has mixed qualities when compared to other traditional crude-based and bio-based polymers. Shellac has a melting point of roughly 75 °C, similar to polycaprolactone (PCL). Because of this lower melting temperature, Shellac is suitable for solid packaging products or cold liquids. In terms of mechanical properties, the tensile strength of shellac film is similar to that of Polyethylene (PE). The elongation at break is only 3.05%, making it unsuitable for flexible packaging applications. However, the literature has shown that the addition of PEG 35,000 increases the elongation at break to 170%, which is better than PET, polylactic acid (PLA), and poly(3-hydroxybutyrate)-co-(4-hydroxybutyrate) (P3HB4HB). Shellac's intrinsic hydrophobicity makes it more competitive with other biopolymers, such as PLA, P3HB4HB, and PCL. The contact angle of Shellac is comparable to that of PE, PET, and polypropylene (PP). The contact angle provides information regarding surface wettability; good water barrier qualities, such as WVPC, are required for the package to preserve the goods. When comparing the WVPC, Shellac has a greater WVPC (lower barrier) than standard PE and PP. However, the barrier is equivalent to PET in other crude-based polymers. Compared to bio-based polymers, Shellac has two times and six times greater barriers than PCL and PLA.

**Table 7.** Comparative analysis of Shellac with other common polymers used in packaging.

| Properties | Shellac | PE | PP | PET | PVC | PCL | PLA | P3HB4HB |
|---|---|---|---|---|---|---|---|---|
| Melting Point (°C) | 75–90 [15] | 115–135 [93,94] | 170 [93] | 245–255 [93,94] | 210 [94] | 68 [95] | 155–165 [95] | 167.59 [96] |
| Tensile Strength (MPa) | 5.7–14 [55] | 7–25 [97] | 27–98 [97] | 157–177 [97] | 42–55 [97] | 38.3 [98] | 37.6 [99] | 87.4 [96] |
| Elongation (%) | 3.05 [56] | 300–900 [97] | 200–1000 [97] | 70 [97] | 20–180 [97] | 839.2 [98] | 59.2 [99] | 28.1 [96] |
| Water Contact Angle (°) | 88.07 [100] | 88 | 88 [101] | 76 [101] | 90 [101] | 80 [98] | 65.2 [99] | 64.7 [96] |
| WVP × $10^{14}$ (gm-m/m2-s-Pa) | 834–1150 [52,58,59] | 6.673–8.704 [97] | 201–401 [97] | 501–1980 [97] | 18.279 [97] | 1680 [98] | 4820 [99] | 359 [96] |

In addition to PCL, PLA, and P3HB4HB, other edible bio-based polymers, such as starch, gelatin, pectin, chitosan, and guar gum, are compared to Shellac in Table 8. In terms of tensile strength, shellac film is better than starch and pectin. However, gelatin, chitosan, and guar gum have higher tensile strength. Shellac, when modified with acrylic, has a similar tensile strength to chitosan but can reach a tensile value similar to guar gum or gelatin. However, Shellac has a similar elongation at break to gelatin but has smaller values than the other edible bio-based polymers mentioned. As shown earlier, adding PEG 35,000 increases the elongation at break to 170%, which is better than starch, gelatin, pectin, chitosan, and guar gum. Shellac's intrinsic hydrophobicity makes it highly competitive with these edible polymers. The contact angle of Shellac is more than the other edible polymers mentioned, and it is worth noting that the WVPC of Shellac is 33 to 44 times lower than guar gum and 8 to 10 times lower than gelatin, showing superior hydrophobicity when compared to all the other edible bio-based polymers mentioned. Therefore, Shellac has untapped potential for packaging applications, including food packaging, due to its non-toxic nature.

**Table 8.** Comparative analysis of Shellac with edible bio-based polymer used in packaging.

| Properties | Shellac | Starch | Gelatin | Pectin | Chitosan | Guar Gum |
|---|---|---|---|---|---|---|
| Tensile Strength (MPa) | 5.7–14 [55] | 2.4 [102] | 57.16 [103] | 3.63 [104] | 14.95 [105] | 18.01 [106] |
| Elongation (%) | 3.05 [56] | 50 [102] | 2.96 [103] | 43.77 [104] | 8.26 [105] | 31.58 [106] |
| Water Contact Angle (°) | 88.07 [100] | 23.18–66.91 [107] | 72 [103] | 31.69 [104] | 52.5 [108] | N.A. |
| WVP × $10^{14}$ (gm-m/m2-s-Pa) | 834–1150 [52,58,59] | 23,000–35,000 [102] | 8890 [103] | 55,300 [104] | 63,400 [109] | 386,100 [110] |

## 10. Conclusions

The word Shellac brings an image of a water-resistant varnish to mind, so it is natural to conjure images of paint and fruit coating when thinking of Shellac. This literature review on Shellac revealed that Shellac is naturally sourced from a lac bug "*kerria lacca*". The usage of Shellac in history dates back to 3000 B.C. It has been used for multiple applications, such as pigment, dyes, varnishes, coating, paint, and it is presently applied in candies and pharmaceutical pills, coating on fruits for shelf life extension, primers, smart sensors, 3D printing, and green electronics. The usage of Shellac in packaging is limited but is now being explored by modifying it to reduce the self-polymerization of Shellac.

Self-polymerization is caused by the inter- and intra-esterification of polyhydroxy carboxylic acids with the free alcohols and is called the aging of Shellac. Significant studies on Shellac involve the retardation or termination of the aging process through either physical or chemical modification. The modification of Shellac to increase flexibility and slow or terminate aging will be helpful for the packaging application process. Moreover, Shellac's unique film-forming ability when solvent cast using ethanol or alkaline solution is like icing on the cake for the packaging application. Many plasticizers and polymers have been blended with Shellac to enhance their properties. The physically modified Shellac showed better elasticity when mixed with most hydrophilic plasticizers or polymers, but a decrease in tensile strength was observed. When modified using acrylates or amines, Shellac showed better tensile strength and elasticity, making it suitable for packaging applications. Conversely, when Shellac was used as reinforcement in other polymers, Shellac helped in increasing the water barrier and tensile properties of the composite film. The literature on Shellac for the packaging application is limited. Still, it has proved its importance in packaging compared to other crude-based polymers, such as PP, PE PET, and PC, or bio-based polymers, such as PCL, PLA, and P3HB-4HB.

In conclusion, Shellac, a renewable raw material with non-toxic and biodegradable properties, can be easily modified for packaging applications by tuning the tensile strength, elongation, and water barrier essential for packaging applications. Therefore, it can be said that shellac-based edible material has untapped potential for packaging applications.

**Author Contributions:** A.A.: Conceptualization, Investigation, Formal analysis, Visualization, Writing—Original Draft. V.K.R.: Conceptualization, Investigation, Resources, Formal analysis, Manuscript editing and review, Supervision. All authors have read and agreed to the published version of the manuscript.

**Funding:** The work was supported by the Prime Minister Research Fellowship (PMRF), Ministry of Education, Govt of India (PM-31-22-677-414).

**Institutional Review Board Statement:** Not applicable.

**Informed Consent Statement:** Not applicable.

**Data Availability Statement:** Not applicable.

**Acknowledgments:** Author Arihant Ahuja would like to thank the Indian Institute of Technology Roorkee and the Prime Minister Research Fellowship for providing online resources and support to carry out this review study.

**Conflicts of Interest:** The authors declare no conflict of interest.

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
