# Peer review of "Shellac: From Isolation to Modification and Its Untapped Potential in the Packaging Application"

_sustainability, doi:10.3390/su15043110_

Round 1

Reviewer 1 Report

The authors presented an original review on shellac. The article is well written, contains a sufficient number of sources, necessary comparative tables and bibliographic analysis.

However, there are a number of minor remarks:

1. The title of the article, one of the functions of packaging is isolation. In this regard, I sit to reconsider the name.

2. Conclusion: the first paragraph is of a general nature, the second paragraph contains a description of the unique features of the material, the third contains generalizing material about the use of shellac in packaging. The structure of the conclusion needs to be made more consistent.

3. The annotation needs similar processing.

After these minor changes, the article can be published.

Author Response

Dear Reviewer, 

Please find our response in the file attached. 

Regards

Dr. Vibhore Rastogi

Reviewer 2 Report

The authors submitted review article is on the "Shellac: From isolation to modification and its untapped potential in the packaging application". The manuscript is well written and can be accepted after few modification.

Abstract: The section needs to modify to show the need of the review article.

Introduction: The section is well written , kindly add more details and comparing Shellac with other Biopolymers, bio-based polymers, and biodegradable polymers to show the advantage the material provide. Various type of biopolymers can be listed in the section with properties.

The extraction and production section can be elaborated and sub-sectioned into different techniques to synthesized Shellac.

Kindly add a different section for chemical properties of Shellac. 

Section 5 and 6: Kindly add more details about the experimental results obtained by authors and images instead citing them in tables.

Conclusion: The section is well written.

The manuscript have some grammatical error which can be corrected in the revised version. 

Author Response

(The authors gave the same response as above.)

Round 2

Reviewer 2 Report

The authors have made significant changes and thus manuscript can be accepted.